# Heterotopic Ossification after a Prolonged Course of COVID-19: A Case Report and Review of the Literature

**Jacob E. Milner \***, **Ean C. Schwartz, Joseph S. Geller, David Constantinescu** , **Paul R. Allegra, Justin E. Trapana**  **and Fernando E. Vilella**

Department of Orthopedic Surgery, University of Miami Miller School of Medicine, Miami, FL 33136, USA
\* Correspondence: jmilner@med.miami.edu; Tel.: +1-(203)-927-7339

**Abstract:** We report the case of a 20-year-old male who developed severe HO of the left hip secondary to a prolonged course of COVID-19 pneumonia. Upon extubation, he was found to have debilitating left hip pain and significant functional deficits with regard to his range of motion and functional status. There are numerous known causes of heterotopic ossification (HO), including trauma, surgery, and traumatic brain or spinal cord injuries. An increased incidence of HO has also been reported in patients who undergo prolonged intubation. While the COVID-19 virus has many known respiratory and medical complications, it has also resulted in unforeseen complications that present long-term challenges for patients. When treating patients with coronavirus, physicians should be aware of HO as a possible complication and consider it as a cause of musculoskeletal pain.

**Keywords:** heterotopic ossification; COVID-19; coronavirus; pneumonia; hip pain; radiograph





## 1. Introduction

Heterotopic ossification (HO), also referred to as myositis ossificans, is a non-neoplastic process involving the development of mature extra-skeletal bone in the muscle or soft tissue [1]. The pathogenesis is thought to be a multifactorial process associated with the release of prostaglandins from soft-tissue mesenchymal stem cells as a response to local inflammation [2]. Although rare genetic causes exist, HO is increasingly recognized to be a systemic inflammatory response to traumatic injuries, surgery, burns, acute respiratory distress, or neurological injury. Comprehensive reviews have shown that up to 75% of cases of HO can be directly attributed to a history of trauma, with "microtrauma" or repeated mechanical stress and systemic inflammation believed to be the underlying aggressors in the remaining cases [3]. HO most frequently presents in the second and third decades of life, with men slightly more affected than women, which is likely related to the fact that young males in this age group are the demographic most likely to sustain traumatic injuries [1].

HO has recently been reported as an atraumatic complication of COVID-19 infections, likely related to the body's systemic inflammatory response to the virus. Meyer et al. (2021) reported on three patients with COVID-19 requiring prolonged intubation, who sustained severe hip HO that required mechanical ventilation [2]. All three patients were 60 years or older [2]. Another case reported the development of HO in a patient with longstanding fibrodysplasia ossificans progressive (FOP). A post-COVID-19 cytokine panel of this patient revealed elevations in the cytokines related to bone remodeling and cortical bone formation as a direct result of acute COVID-19 infection, suggesting a relation between the inflammatory processes in the context of COVID-19 infection and ectopic bone formation [4,5].

We present the case of a 20-year-old male who developed bridging HO of the hip after a complex hospital course due to COVID-19. To our knowledge, this is the first case report of COVID-19-induced symptomatic heterotopic ossification in a young, previously healthy

individual. As a result, clinicians should be aware that this complication can also occur in young patients.

## 2. Case Report

A 20-year-old male was admitted to the hospital with worsening hypoxemic respiratory failure due to COVID-19 pneumonia. His past medical history was significant only in terms of obesity, with no other comorbidities or substantial family history. He was initially admitted to the intermediate medical care unit (IMCU) but was transferred to the critical care unit (CCU) ten days later after requiring intubation and mechanical ventilation secondary to respiratory distress. Approximately 16 days after intubation, the patient developed a tension pneumothorax after the attempted repositioning of his endotracheal tube, and he subsequently sustained pulseless electrical activity and cardiac arrest. Cardiopulmonary resuscitation was performed, and the return of spontaneous circulation was achieved. The patient's hospital course was further complicated by pseudomonas bacteremia, for which he completed a 7-day course of cefepime. He experienced no other significant complications and was extubated for a total of 69 days. His hospital length of stay (LOS) was 90 days.

Approximately two months after admission and 49 days after intubation, the patient underwent an abdominal CT scan indicated for the evaluation of a potential ileus. At that time, HO of the left hip was incidentally found on CT scan. A physical exam and the patient's reporting of symptoms were impossible at that time due to intubation and sedation.

Upon extubation and further evaluation, the patent complained of groin pain upon movement and was unable to ambulate with physical therapy. Left hip radiographs were obtained, revealing Brooker stage III bridging heterotopic ossification (Figure 1, Table 1) [6]. On examination of the left lower extremity, he was found to have significant deficits in hip flexion, extension, and abduction. Nearly all active and passive ranges of motion caused him discomfort and pain. As a result, the patient was unable to ambulate independently. He was neurologically intact distally.

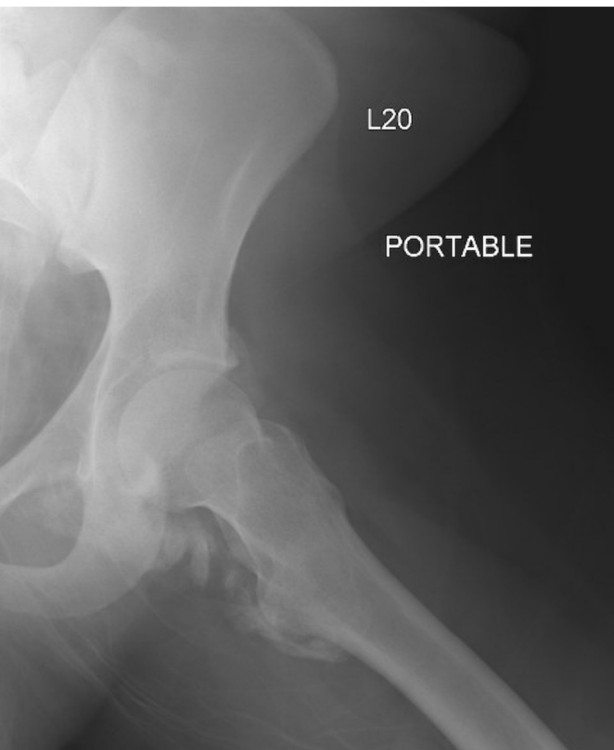

**Figure 1.** Brooker Stage III Heterotopic Ossification of the Left Hip, Seen After an Approximately Two-Month Course of COVID Pneumonia Requiring Prolonged Intubation.

**Table 1.** Brooker Classification of Heterotopic Ossification Around the Hip Joint [6].

| Stage | Description |
| --- | --- |
| I | Bone islands within the soft tissues |
| II | Bone spurs from the pelvis or proximal end of the femur, with at least 1 cm between opposing bone surfaces |
| III | Bone spurs from the pelvis and/or proximal end of femur, with <1 cm between opposing bone surfaces |
| IV | Apparent bone ankylosis of the hip |

## 3. Discussion

Heterotopic ossification is a known complication of severe illness, resulting in prolonged systemic inflammation. However, it most commonly occurs as secondary to high-energy trauma or neurologic injury. Studies of central nervous system (CNS) injuries and/or neuromuscular paralysis suggest that muscle denervation may lead to the development of HO, with the prevalence of HO following CNS injury ranging from 10% to 53% [3,7]. Furthermore, therapeutic neuromuscular blockade for the management of respiratory failure has been shown to lead to the development of HO, also likely to be secondary to muscular denervation [8]. The pathophysiology is believed to be related to local circulatory, metabolic, and biochemical changes as a result of prolonged muscle denervation, leading to ectopic bone formation in the soft tissue [8,9]. As such, heterotopic ossification must be considered in the case of critically ill patients who require mechanical intubation and neuromuscular paralysis.

Within the scope of COVID-19, prolonged immobilization and neuromuscular paralysis in the context of mechanical ventilation may play a role in the development of HO. However, additional factors, such as deranged calcium metabolism, local tissue hypoxia, and a systemic inflammatory response secondary to COVID-19, likely also contribute to the development of HO [10,11]. Reviews of acute COVID infections have shown substantial innate and adaptive immune responses that contribute to the inflammatory response that ultimately results in critical illness. The immunological profiles of patients with severe COVID-19 infection suggest an increased inflammatory response compared to asymptomatic patients, with increased serum inflammatory markers, such as C-reactive protein (CRP), lactate dehydrogenase (LDH), and IL-6 [12].

Specifically, high levels of IL-6 in patients with acute COVID-19 infection are associated with a poor prognosis and even death, and they are more likely to be reduced compared to the peak levels in patients who show clinical improvement [13,14]. A recent meta-analysis of 16 studies that included 10,798 Chinese patients with COVID-19 infection showed elevated IL-6 levels in all COVID-positive patients, with a 2.9-fold increase in the serum levels of patients with severe infection [15]. Thus, a strong relationship exists between elevated inflammatory markers and severe COVID infection, which may be related to the inflammatory response that results in HO. With regard to HO, inflammatory modulators have been studied as potential mitigators of ectopic bone formation. Specifically, BMP-1 receptor inhibitors have been shown to reduce the IL-6 levels and subsequent bone formation, suggesting the role of IL-6 cytokines as a contributor to HO formation [16].

A review of the literature reveals eight individually reported cases of HO in the context of COVID-19 infection. Of these, four patients were female and four were male, with an average age of 51.5 years old (39–74). Seven of the eight (87.5%) patients who developed HO required mechanical ventilation for an average of 27.8 days. All eight patients who developed HO in the context of COVID-19 infection had multiple comorbidities, with the most common being hypertension (*n* = 5) and COPD (*n* = 2). Three patients developed hip HO, three developed elbow HO, one developed knee HO, and one was found to have HO of the neck (Table 2) [2,4,11,17]. An additional literature review revealed a study of 52 patients with severe COVID-19 requiring intubation and mechanical ventilation, 10 of whom developed HO [18]. These patients presented with a mean age of 71 and required

longer mechanical ventilation (36 days) than those who did not develop significant HO (22 days) ($p < 0.001$) [18].

**Table 2.** Demographic Information from All Case Reports of Described COVID-19-Induced Heterotopic Ossification in the Literature.

| Patient | Age | Gender | LOS (Days) | Mechanical Ventilation (Days) | Comorbidities | HO Location | Laterality |
|---------|-----|--------|-----------|------------------------------|---------------|-------------|------------|
| 1 | 64 | Male | - | 26 | HTN, Afib | Hip | Bilateral |
| 2 | 73 | Male | - | 27 | HTN, COPD | Hip | Left |
| 3 | 74 | Male | - | 30 | HTN, COPD | Hip | Left |
| 4 | 39 | Male | - | 28 | Schizophrenia, Bipolar, Etoh abuse | Shoulder | Bilateral |
| 5 | 51 | Female | 47 | - | HTN, T2DM | Shoulder | Bilateral |
| 6 | 43 | Female | 33 | - | HTN | Shoulder | Right |
| 7 | 45 | Female | 0 | 0 | FOP | Neck | Left |
| 8 | 23 | Female | 81 | - | Postpartum | Knee | Left |

In light of the ongoing pandemic and increased number of hospitalizations requiring mechanical intubation, the early recognition and proper management of heterotopic ossification is an important consideration for obtaining improved patient outcomes. Early symptoms of HO include localized pain, tenderness, and swelling, with the later stages showing a restricted range of motion and soft tissue stiffness [1]. However, these symptoms may be difficult to assess in patients who are intubated and/or sedated. Radiography is most often the first imaging technique used to identify HO in symptomatic patients, although incidental findings on CT scan can also reveal HO, as seen in the present case report. HO can lead to severe functional impairment. Thus, early identification, combined with well-executed treatment strategies, is imperative to the ultimate improvement of patient outcomes [3]. As previously mentioned, this is a challenge in the case of intubated and/or sedated patients, as well as patients with traumatic brain injuries.

The prophylaxis and treatment of HO are often multifaceted and vary depending on the joint involved, as well as the patient's symptoms and functional limitations. NSAIDs have been proposed as a prophylactic method for use during hospitalization that work by inhibiting the formation of Prostaglandin-E2, a major contributor to the formation of ectopic bone tissue. Specifically, indomethacin and Celecoxib have been shown to be effective in preventing clinically significant HO following total hip arthroplasty and after acetabular fracture [19]. A small dose of radiation therapy has also been shown to be effective in preventing clinically significant HO of the hip after acetabular fractures [20]. While multiple forms of prophylaxis have been shown to prevent hip HO after trauma, the evidence is not as strong as that for other joints with respect to the prevention of HO. However, a recent study by Geller et al. (2020) suggests that prophylactic radiotherapy prevents clinically significant elbow HO and decreases the need for future surgery [21]. Prophylactic radiotherapy, however, is likely not an option in the absence of trauma, as systemic inflammation cannot be targeted with radiation therapy for a single joint.

Bisphosphonates, which induce osteoclast apoptosis and inhibit calcification, have also been proposed as a possible prophylactic measure [22]. Previous analyses have revealed that prophylactic NSAID administration in spinal cord injury (SCI) patients has been effective, while bisphosphonate therapy showed superior efficacy in the treatment of established HO in SCI patients [22]. As such, clinicians may want to consider prophylactic NSAIDs and/or bisphosphonate therapy for HO prevention in high-risk patients with COVID-19 requiring mechanical intubation. Once clinically significant HO has formed, the treatment options are essentially limited to physical therapy and surgical excision of the ectopic bone [3]. Surgery, while often necessary in order to alleviate debilitating symptoms,

can lead to numerous complications, including neurovascular injury, infection, and even the recurrence of heterotopic bone. Therefore, it is much more beneficial to prevent HO than to treat it after it has already formed. Consequently, the prophylactic management of HO in high-risk patients should be considered in the context of COVID-19 requiring mechanical ventilation.

## 4. Conclusions

The pathophysiology of heterotopic ossification is related to several risk factors, including patient demographics, the mechanism of injury, systemic inflammation, the presence of a neurologic injury, and the length of mechanical ventilation. Heterotopic ossification secondary to COVID-19 has been sparingly described in the literature and has predominantly been found in older patients with numerous medical comorbidities.

In this case report, we described debilitating heterotopic ossification of the left hip in a 20-year-old male, representing the youngest patient to develop heterotopic ossification secondary to coronavirus described in the literature to date. This case is an important contribution to the current literature on HO development in the context of COVID-19. Healthcare providers must be aware of heterotopic ossification as a potentially devastating complication after a prolonged course of COVID pneumonia and should consider prophylactic treatment for high-risk patients. Our case report is limited due to a lack of adequate patient follow up. The patient was lost to follow up and, therefore, this case does not provide information regarding further treatments, medications, further evaluation, or surgical treatments. Additionally, further prospective studies with larger cohorts are necessary in order to determine the incidence and functional impact of HO on hospitalized patients with COVID-19.

**Author Contributions:** Conceptualization, J.E.M., J.S.G. and E.C.S.; methodology, J.S.G., D.C., P.R.A. and J.E.T.; formal analysis, J.E.M., J.S.G. and E.C.S.; investigation, J.E.M. and E.C.S.; resources, J.S.G., D.C., P.R.A. and J.E.T.; data curation, J.E.M. and E.C.S.; writing—original draft preparation, J.E.M. and E.C.S.; writing—review and editing, J.S.G., D.C., P.R.A., J.E.T. and F.E.V.; visualization, J.E.M., J.S.G. and E.C.S.; supervision, F.E.V.; project administration, F.E.V. All authors have read and agreed to the published version of the manuscript.

**Funding:** This research received no external funding.

**Institutional Review Board Statement:** Patient's verbal consent was obtained prior to patient discharge from the hospital. Due to loss to follow up there was no written consent obtained.

**Informed Consent Statement:** Verbal consent was obtained from this patient. Written consent was deferred due to loss to follow up.

**Data Availability Statement:** Not applicable.

**Conflicts of Interest:** The authors declare no conflict of interest.

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
