# Peer review of "Heterotopic Ossification after a Prolonged Course of COVID-19: A Case Report and Review of the Literature"

_traumacare, doi:10.3390/traumacare2040045_

Round 1

Reviewer 1 Report

First of all, You have reported an interesting case that you think is related to COVID-19.

The overall design of the report is not well thought out.

1) Please provide a reference to Meyer et al. on line 37 in the first mentioned sentence.

2) Is "all age" suggested in line 49 been reported by various age groups so far? If there is no evidence, please correct it to "We should be aware that it can also occur in young patients".

3) Mark Fig.1 on line 70.

4) In line 71, please present the Brooker stage as a table to help readers understand.

5) Line 104: Separately describe the result and discussion on line 104. Since then, report on the patient's care and progress.

6) It is not a care report to simply report an unusual matter. After a certain time has elapsed, the patient's condition and results should also be reported. Please add what kind of medication you are taking, what kind of further evaluation you are doing, and whether it is accompanied by surgical treatment.

Author Response

Reviewer #1,

Thank you so much for your feedback on our paper and providing your suggestions. Below I have explained the revisions made for each point:

1) Please provide a reference to Meyer et al. on line 37 in the first mentioned sentence.

Reference was added

2) Is "all age" suggested in line 49 been reported by various age groups so far? If there is no evidence, please correct it to "We should be aware that it can also occur in young patients".

Revisions were made to say "As a result, clinicians should be aware that this complication can also occur in young patients"

3) Mark Fig.1 on line 70.

Marked figure 1.

4) In line 71, please present the Brooker stage as a table to help readers understand.

Please see the added table 1 describing the Brooker staging with references attached

5) Line 104: Separately describe the result and discussion on line 104. Since then, report on the patient's care and progress.

Per your suggestion and reviewer #2's suggestions, we changed this heading just to "discussion"

6) It is not a care report to simply report an unusual matter. After a certain time has elapsed, the patient's condition and results should also be reported. Please add what kind of medication you are taking, what kind of further evaluation you are doing, and whether it is accompanied by surgical treatment.

This was a point of ours that we were sure to make. We added additional comments in the conclusion that addressed the lack of patient follow up as a limitation to this case report. We were unable to attain adequate follow up and as such had to report this as a limitation of ours.

Thank you once again for the recommendations. We appreciate you taking the time to review our work and providing feedback!

Th

Reviewer 2 Report

Thank you for the opportunity to review this article.
This is a case report of an HO case in patient with Covid-19.
The topic is very interesting as I think there are many similar cases that occurred and have not yet been described. We also had a similar case in my Institute. Covid-19 could be a relevant risk factor. Therefore, I find the presentation of these cases in the literature useful.
I think the writing and methodology are good. The introduction provides all the elements useful for understanding, the case is adequately described, and the discussion is interesting.
I think the article is worthy of publication after minor revisions. In particular, I would advise the authors to revise the first sentence of the introduction, which I find incorrect. I recommend reading the following article:
"Meyers C, Lisiecki J, Miller S, Levin A, Fayad L, Ding C, Sono T, McCarthy E, Levi B, James AW. Heterotopic Ossification: A Comprehensive Review. JBMR Plus. 2019 Feb 27;3(4):e10172. doi: 10.1002/jbm4.10172. PMID: 31044187; PMCID: PMC6478587."
Also I would rename paragraph "3" just "Discussion".
Thank you.

Author Response

Thank you so much for reviewing our work and providing feedback!

Please see the responses to your suggestions below:

  1. I think the article is worthy of publication after minor revisions. In particular, I would advise the authors to revise the first sentence of the introduction, which I find incorrect. I recommend reading the following article:"Meyers C, Lisiecki J, Miller S, Levin A, Fayad L, Ding C, Sono T, McCarthy E, Levi B, James AW. Heterotopic Ossification: A Comprehensive Review. JBMR Plus. 2019 Feb 27;3(4):e10172. doi: 10.1002/jbm4.10172. PMID: 31044187; PMCID: PMC6478587."
This article was referenced as the first paper in our references section. I revised the initial sentence per your recommendations, which now reads: " Heterotopic ossification (HO), also referred to as myositis ossificans, is a non-neoplastic process involving the development of mature extra skeletal bone in muscle or soft tissue."

2. Also I would rename paragraph "3" just "Discussion".

Paragraph 3 has been renamed to just "discussion"

Thank you so much again for taking the time to review our paper and provide your feedback!

Reviewer 3 Report

The patient history is unclear. The letter describes timing of finding for the HO at 2 months following admission, 49 days after intubation - suggesting intubation without admission?   

It describes the HO as an incidental finding in a CT scan but it is unclear if this is at 2 months or 45 days?   Then it seems to suggest a second "finding" of HO at extubation when the patient complained of pain.  Please revise to more clearly describe the course of events.

Authors introduction and discussion are missing several studies/case reports of covid-19 and  HO, including those published in 2020 and 2021.  The literature review should be updated.  In particular authors should consider including the following reference as it also indicates symptomatic patients: 

Stoira E, Elzi L, Puligheddu C, Garibaldi R, Voinea C, Chiesa AF; Collaborators. High prevalence of heterotopic ossification in critically ill patients with severe COVID-19. Clin Microbiol Infect. 2021 Jul;27(7):1049-1050. doi: 10.1016/j.cmi.2020.12.037. Epub 2021 Jan 15. PMID: 33460831; PMCID: PMC7833636.

and this, as it describes HO in a young patient.

Liu J, Luther L, Dwivedi S, Evans AR. Long-term Orthopedic Manifestations of COVID-19: Heterotopic Ossification and Digital Necrosis. R I Med J (2013). 2022 Sep 1;105(7):31-35. PMID: 35930488.

Author Response

Thank you so much for your feedback and for providing us with suggestions on how to improve our paper. Please see the responses below:

  1. The patient history is unclear. The letter describes timing of finding for the HO at 2 months following admission, 49 days after intubation - suggesting intubation without admission?  

The timing of this describes that our patient's HO was discovered around 2 months following admission (roughly 60 days following admission). On day 10, the patient was intubated. As such, the HO was discovered around 49 days following intubation.

2. It describes the HO as an incidental finding in a CT scan but it is unclear if this is at 2 months or 45 days?   Then it seems to suggest a second "finding" of HO at extubation when the patient complained of pain.  Please revise to more clearly describe the course of events.

Please see the revised wording the more clearly describe the course of events: 

Approximately two months after admission and 49 days after intubation, the patient underwent abdominal CT scan indicated for evaluation of a potential ileus. At that time, HO of the left hip was incidentally found on CT scan. Physical exam and patient reported symptoms were deferred at that time due to intubation and sedation.

Upon extubation and further evaluation, the patent complained of groin pain with movement and was unable to ambulate with physical therapy. Left hip radiographs were obtained revealing Brooker stage III bridging heterotopic ossification [Figure 1][Table 1] 6

3. Authors introduction and discussion are missing several studies/case reports of covid-19 and  HO, including those published in 2020 and 2021.  The literature review should be updated.  In particular authors should consider including the following reference as it also indicates symptomatic patients: 

Stoira E, Elzi L, Puligheddu C, Garibaldi R, Voinea C, Chiesa AF; Collaborators. High prevalence of heterotopic ossification in critically ill patients with severe COVID-19. Clin Microbiol Infect. 2021 Jul;27(7):1049-1050. doi: 10.1016/j.cmi.2020.12.037. Epub 2021 Jan 15. PMID: 33460831; PMCID: PMC7833636.

and this, as it describes HO in a young patient.

Liu J, Luther L, Dwivedi S, Evans AR. Long-term Orthopedic Manifestations of COVID-19: Heterotopic Ossification and Digital Necrosis. R I Med J (2013). 2022 Sep 1;105(7):31-35. PMID: 35930488.

Please see the updated discussion section which includes a reference to the Stoira paper, as well as an updated table 2 to include the case report described by Liu et. al

Thank you

Thank you so much again for your suggestions!

Round 2

Reviewer 1 Report

As mentioned earlier, mentioning only certain situations is not valuable as a case report. Patient follow-up is essential. Otherwise, it is not suitable as paper.

Reviewer 2 Report

The authors have addressed all my concerns. I consider the article suitable for publication in its current form. Thank you.